# Knowledge, Opinions, and Practices of Oral Cancer Prevention among Palestinian Practicing Dentists: An Online Cross-Sectional Questionnaire

**DOI:** 10.3390/healthcare11071005

**Published:** 2023-03-31

**Authors:** Rola Muhammed Shadid, Ghassan Habash

**Affiliations:** 1Department of Prosthodontics, Faculty of Dentistry, Arab American University, Jenin P.O. Box 240, Palestine; 2Private Practice, Tulkarm P.O. Box 304, Palestine; 3Department of Dental Sciences, Faculty of Graduate Studies, Arab American University, Ramallah P.O. Box 240, Palestine; 4Palestinian Dental Association, Jerusalem P.O. Box 19183, Palestine

**Keywords:** practicing dentist, Palestine, opinion, oral cancer, knowledge, mucosal screening

## Abstract

Oral-health professionals play a critical role in the early recognition and prevention of oral cancer; however, most patients are diagnosed in the advanced stages that demonstrate poor prognosis. Therefore, this study aimed to investigate the knowledge, opinions, and clinical practices of practicing dentists related to oral cancer prevention and oral mucosal screening in Palestine. This research used an online cross-sectional questionnaire design. Practicing dentists in Palestine completed the 44-item questionnaire encompassing the following four sections: (a) personal data (6 items); (b) oral cancer knowledge (21 items); (c) opinions and beliefs related to oral cancer (10 items); and (d) clinical practices related to oral cancer (7 items). The questionnaire was sent to all eligible participants (*N* = 650) between July and September of 2022. The response rate was 39.1% (*N* = 254). About 70.1% of the respondents had poor knowledge of oral cancer and its risk factors, and almost 84.3% had positive opinions related to oral cancer prevention and oral mucosal screening. Concerning the clinical practices, only 29.9% routinely examined their patients every time their patients visited. More experienced dentists showed significantly more positive opinions and better clinical practices related to oral cancer prevention and screening; however, they had significantly lower knowledge scores compared with the younger dentists (*p* < 0.05). Inadequate training was the most reported perceived boundary against routine oral cancer screening. Palestinian dentists appeared to have positive opinions related to oral cancer prevention and oral mucosal screening. However, the assessment of the knowledge and clinical practices revealed deficiencies in this regard.

## 1. Introduction

Oral cancer denotes all malignant tumors that affect the tissues of the oral cavity, including the vermillion border of the lips, the buccal mucosa, the tongue, the floor of the mouth, and the palate [1]. Oral cancers emanate mainly from epithelium cells, and approximately 90% are squamous cell carcinomas [1]. Oral cancer is the 16th most common cancer internationally, with around 377,713 new diagnoses and almost 177,757 deaths reported in 2020 [2]. However, the incidence and fatality rates of oral cancer are higher in low- and middle-income transitioning countries [3].

In Palestine, as a low-income transitioning country, new diagnoses of oral cavities and oropharyngeal cancer were reported to be around 90 and the associated deaths were 35 in 2012 [4]. The Palestinian Ministry of Health has taken several steps to address this issue, including public awareness campaigns, training of healthcare professionals in early detection and diagnosis, and the establishment of cancer treatment centers. These efforts aim to reduce the burden of oral cancer in Palestine and to improve the overall health outcomes of the population. However, more efforts should be made to document the new cases each year, to control the risk factors, and to increase awareness among healthcare professionals and population.

Oral cancer is a multifactorial disease [5], but the main risk factors include oral tobacco consumption, excess alcohol drinking, human papilloma virus related to cancers in the oropharyngeal regions, genetic vulnerability [5,6,7,8], and sun exposure of ultraviolet radiation related to lip cancer [9]. There are also other risk factors related to the environment, diet and occupation [10]. However, the endemic oral tobacco consumption is rendered the major risk factor in Palestine; since 40% of the Palestinian male population are smokers [10].

Oral cancers generally reveal promising survival rates if diagnosed in the early stages [11]; unfortunately, most patients are diagnosed in the advanced stages (III and IV) [12]. The 5-year survival rates of stages III and IV are reported to be 41% and 9%, respectively; whereas stages I and II show 85% and 66% 5-year survival rates, respectively [13].

In an effort to reduce the incidence and mortality of oral cancer, the World Health Organization (WHO) Global Oral Health report written by a group of experts has endorsed the following two methods to prevent oral cancer: lessening exposure to risk factors and screening for oral potentially malignant lesions [14]. Responding to both preventive measures, high-income countries have witnessed a dramatic shift in the forms and tendencies of oral cancer over the past 30 years [15]. The incidence and mortality rates of oral cancer are estimated to duplicate by 2030 in transitioning countries in the Middle East and North Africa, and to rise by only 50% in the same time span worldwide [16]. Therefore, unremitting efforts should be made to decrease the incidence and mortality of oral cancers internationally, with a distinct focus on transitioning regions.

Since oral cancers are usually anteceded by clinically discernible mucosal changes, recognized as premalignant or potentially malignant oral lesions [17], oral-health professionals could play a critical role in the early recognition and prevention of oral cancer by routine oral mucosal screening [18].

Several studies worldwide have assessed oral-health professionals’ knowledge, beliefs and clinical practices concerning oral mucosal screening and oral cancer prevention [18,19,20,21,22,23,24,25,26,27,28,29,30,31,32]. Some of those studies revealed inadequate levels of knowledge or practices of oral cancer prevention [19,20,23,26,27,29,32], suggesting the need for more education programs and improving the curricula of oral cancer in undergraduate and graduate dental courses.

As Palestine is considered to be part of the low-income transitioning region with an increasing trend of oral cancer progression [16], and owing to the crucial role of practicing dentists in reducing the incidence and mortality of oral cancer, this study was conducted. The primary objective of this cross-sectional questionnaire was to investigate the knowledge, opinions, and clinical practices of the practicing dentists related to oral cancer prevention and oral mucosal screening in Palestine. The secondary objectives were to recognize the factors that influence their practices of oral cancer screening, and to assess the effect of sex, years of clinical experience, and work setup on the level of the dentists’ knowledge, opinions, and clinical practice.

The hypothesis was that practicing dentists in Palestine would have satisfactory knowledge, opinions, and clinical practices related to oral cancer prevention and oral mucosal screening.

## 2. Materials and Methods

This cross-sectional study was conducted among the dentists practicing in Palestine between July and September of 2022. The study protocol was approved by the Institutional Review Board, Faculty of Dentistry, Arab American University (2022/A/8/N). It was conducted in accordance with the Declaration of Helsinki standards and in agreement with CHERRIES guidelines [33]. All actively practicing dentists in Palestine, including general practitioners and specialists, were invited to participate in this study. The questionnaire was formed online utilizing Google Forms, and it was shared with all eligible participants (*n* = 650) by using dentistry-associated closed groups on social media and by sending the questionnaire to the eligible dentists individually. The questionnaire was accompanied with a cover letter explaining the purposes of the study, their voluntary and anonymous participation, and the confidential use of the gained information. All participants were asked to provide informed consent.

This cross-sectional study comprised 44 closed-ended questions that were formed from a mixture of questionnaires from previous studies [23,24,32,34]. It was originally prepared in English, then translated into Arabic. The questionnaire was evaluated for content validity by an expert in the field of oral medicine, and it was pilot tested by sending it to a group of 20 dentists to assess the questions’ simplicity and clarity. Subsequently, it was slightly revised by rephrasing a few sentences to make them clearer.

The 44 close-ended questions were assorted into the following 4 sectors: (a) personal data, (b) knowledge of oral cancer risk factors, signs, symptoms, and diagnostic procedures, (c) opinions, and (d) clinical practices related to oral cancer prevention and oral mucosal screening.

For knowledge level assessment, twenty questions were asked. Every correct answer gained a score of “1”, so the overall knowledge score ranged from 0 to 20. The study used (12/20) 60% as the cut-off point [31], while more than 12 points suggested the practitioners had good knowledge and ≤12 points suggested poor knowledge of oral cancer risk factors, signs and symptoms.

For the dentists’ opinions and beliefs, ‘strongly agree’, ‘agree’, ‘disagree’, and ‘strongly disagree’ response formats were collected. A score of 1 indicated a positive opinion, while a score of 0 indicated a negative opinion. Six was set as the cut-off point (60%; 6/10); more than 6 counted as a positive opinion and ≤6 as a negative opinion related to oral cancer prevention and oral mucosal screening.

Regarding the clinical practices of oral cancer prevention and oral mucosal screening, six questions were asked, and another item was about the claimed dentists’ boundaries against routine oral mucosal screening.

Reliability was evaluated by means of the test–retest method in which 20 dentists filled out the questionnaire twice within two weeks. By comparing the results of the two times, the Pearson’s correlation coefficient revealed significant stability, indicating good test–retest reliability. Internal consistency that reflected the inter-correlation between the items in the questionnaire was quantified using the coefficient “Cronbach’s alpha”. A Cronbach α = 0.703 was obtained, suggesting adequate internal consistency [35].

## 3. Statistical Analysis

Responses were gathered using the Google Sheet, and data were statistically analyzed using the Statistical Package for Social Sciences (SPSS), version 22.0. Descriptive statistics of the means, standard deviations (SD), and percentages were calculated for all continuous variables.

The effect of sex on the level of the dentists’ knowledge, opinions, and clinical practice was assessed using the independent samples *t*-test, and the influence of years of experience and work setup was tested using one way analysis of variance (ANOVA). The level of statistical significance was set at *p* < 0.05.

## 4. Results

Of the 650 dentists who were invited to participate, 254 completed the questionnaire, representing a response rate of 39.1%. Of those respondents, 56.3% were men, 79.9% were general dental practitioners, and about 80.7% worked in private practices. Approximately one third of the respondents were recent graduates of less than 5 years of clinical experience, whereas nearly one quarter of the respondents had more than 15 years of clinical experience (Table 1).

With regard to the level of knowledge of oral cancer and its risk factors, about 70.1% of the participants had poor knowledge, with an overall mean of 11.14 ± 2.71. (Table 2 and Table 3). The statistical analysis revealed that there was a significant association between the participating dentists’ knowledge score and the years of clinical experience. Dentists with less than 5 years of clinical experience were significantly more likely to have better knowledge of the diagnosis and risk factors of oral cancer compared with the more experienced dentists (*p* < 0.05). However, dentists’ sex or work setup did not have any significant influence (*p* > 0.05).

Regarding dentists’ opinions and beliefs, the majority of the respondents (84.3%) had positive opinions related to oral cancer prevention and oral mucosal screening, with an overall mean of 7.94 ± 1.49 (Table 3 and Table 4). More experienced dentists of more than 15 years of clinical experience showed significantly more positive opinions regarding oral cancer prevention and screening in comparison with those of less clinical experience (*p* < 0.05). However, dentist’s sex or work setup did not have any significant influence (*p* > 0.05).

Concerning the clinical practices of oral cancer prevention and oral mucosal screening, only 29.9% examined their patients routinely every time their patients visited the clinic. In addition, 17.7% reported that they examined new patients in their first visit, while 43.3% reported that they examined their patients for mucosal lesions only when they had suspicions towards patients with a high risk of oral cancer. Regarding the respondents’ experiences with suspicious oral mucosal lesions, a considerable number (67.3%) reported that they had noticed a suspicious oral mucosal lesion, and 74.4% had referred patients with a suspicious lesion to a specialist. Most dentists refer only to oral surgeons, oral pathologists, or specialists in oral medicine (79.2%), while others refer to either specialists in oral surgery, oral pathology or oral medicine. The majority (80.3%) reported that they offered smoking cessation advice to their patients, whereas only 29.1% used diagnostic aids to help in detecting suspicious oral mucosal lesions. More experienced dentists of more than 15 years of experience showed significantly better clinical practice regarding oral cancer prevention and screening in comparison with those with less clinical experience (*p* < 0.05). However, dentists’ sex or work setup did not have any significant influence (*p* > 0.05).

Concerning the perceived boundaries against routine oral mucosal screening, greater than half of the participants (55.1%) agreed that inadequate training was a factor. Other perceived boundaries encompassed the inadequacy of time (29.3%), inadequacy of confidence (8.8%), lack of effectiveness (4.1%), and shortage of financial incentives (2.7%).

## 5. Discussion

The purposes of this study were to investigate the knowledge, opinions, and clinical practices of practicing dentists related to oral cancer prevention and oral mucosal screening in Palestine; and to recognize the factors that influence their practices of oral cancer screening. The hypothesis stating that practicing dentists in Palestine would have satisfactory knowledge and clinical practice related to oral cancer prevention and oral mucosal screening was rejected. However, it was accepted regarding dentists’ opinions, as most practicing dentists in Palestine revealed good motivation to conduct oral mucosal screening and oral cancer prevention.

There was an 39.1% response rate, which is comparable to other analogous studies in Malaysia (41.7%) [36] and Sri Lanka (38%) [37], and is higher compared to another similar study from Yemen, which reported a 27.6% response rate [18].

Since dentists play a crucial role in oral mucosal screening and oral cancer prevention, they should be knowledgeable in oral cancer risk factors, signs, symptoms, and diagnostic procedures. However, the majority of the respondents in the current study showed a poor level of knowledge related to oral cancer and its associated risk factors. This finding generally agrees with other international studies that showed deficiencies in the oral cancer knowledge of the practicing dentists [19,20,23,26,27,29]. Furthermore, a recent study that assessed oral cancer knowledge of undergraduate dental students and interns in a Palestinian dental school supports the findings of our study, stressing the significant need for education improvement in this regard [32].

Concerning the clinical practices of oral cancer prevention and oral mucosal screening, only 29.9% screened their patients routinely for mucosal lesions, which is higher than that reported in a recent study from Saudi Arabia [38], where only 13% reported routine screening of their patients. However, higher percentages were reported in previous studies from Australia [24], Sudan [28], and Kuwait [30].

Oral cancers are mostly associated with a promising prognosis if detected early [11]. Early detection and excision of potentially malignant oral lesions may prohibit malignant transformation; in addition, the recognition of malignancy in the early stages might improve survival rates [39]. About three quarters of dentists in this study have detected and referred patients with suspicious mucosal lesions to a specialist, such as an oral surgeon, an oral pathologist, or an oral medicine specialist. This fraction is higher compared to other studies from the US [21] and Kuwait [30]. According to the nature of the lesion and the stage at which it is detected, oral medicine or oral surgery specialists with specialized training are the most appropriate health professionals to examine these patients [24]. Furthermore, the present study revealed that nearly three quarters of the surveyed dentists feel comfortable and confident in discussing the presence of a suspicious oral mucosal lesion with their patients. This percentage is lower than that reported for Australian dentists [24], where 91.7% reported feeling comfortable in communicating with their patients regarding mucosal pathology.

Since potentially malignant oral lesions can imitate traumatic lesions that can usually be resolved within two weeks, it is highly recommended that dentists follow up the patient with a suspicious mucosal lesion two weeks following the initial presentation prior to referring them to a specialist [40,41]. However, once the dentist arranges a referral to a specialist for a suspicious lesion, he/she should strongly advise the patient to attend the specialist’s appointment punctually without delay [42]. Positively, the vast majority of dentists in the current study believed in patient follow-up referrals, similar to an analogous previous study [24].

The conventional protocol for oral mucosal screening is through visual and tactile examination; however, diagnostic aids and instruments are introduced to aid in the visualization and detection of malignant and potentially malignant oral lesions [43]. About 29% of the dentists in the current study reported using diagnostic aids for the detection of oral lesions, which is much higher than that reported in a previous study (5.6%) [24].

While most dentists surveyed in this study believed that oral mucosal screening is their duty, 60.6% believed they had sufficient knowledge and training to do so; comparable to the results reported from Sudan [28]. In addition, nearly three quarters of the dentists believed that all patients, not only those at high risk, should be screened, and a comparable percentage did not believe in patients’ ability to detect mucosal pathology on their own. A higher percentage above 90% was reported in a previous study for dentists who believed in screening all patients and did not believe in patients’ self-awareness of oral mucosal pathology [24].

Why dentists may not perform routine oral mucosal screening of their patients could be attributed to various boundaries [24]. In the current study, inadequate training was the most prevalent factor, followed by the inadequacy of time; while inadequacy of confidence, lack of effectiveness and shortage of financial incentive were minorly reported as barriers. The most dominantly reported factor, inadequate training, could be addressed through conducting hands-on continuous education courses and through improving the undergraduate/graduate curricula. On the other hand, the claimed boundary of inadequate time could be addressed by frequent reminding of the practicing dentists of their duty in performing comprehensive head and neck examination for each patient [44]. The dentists should intentionally reserve specific time in a patient’s appointment for comprehensive head and neck examination that may take only about 2 min [24].

Heavy tobacco smoking and alcohol consumption are regarded as two of the major risk factors for oral cancer [5,6,7]. Positively, the vast majority of dentists surveyed in this study believed that they should provide smoking and/or alcohol cessation advice to their patients, and they reported that they did so. This is higher than that reported among Sudanese dentists [28]. However, nearly half of the surveyed dentists believed that they could influence a patient to quit or reduce smoking, concurring with the results reported among Australian dentists [24].

Regarding the impact of years of clinical experience on knowledge, beliefs, and practices of oral cancer prevention, the findings of the current study revealed a significant association between the dentists’ knowledge of oral cancer and years of clinical experience. Recent graduates of less than 5 years of experience had significantly better knowledge of the diagnosis and the risk factors of oral cancer, similar to that reported in previous studies [23,28]. This raises the need for older dentists to attend more continuous educational programs in this regard. On the other hand, more experienced dentists of more than 15 years of clinical experience showed significantly better clinical practice and more favorable opinions regarding oral cancer prevention and screening. This agrees with the results of a previous study [24] that reported that more experienced dentists could talk about the existence of a suspicious oral lesion with their patients more comfortably and frankly, due to the enhancement of their communication skills.

A recent systematic review has shown that potentially malignant oral lesions and malignant lesions are commonly overlooked by oral healthcare practitioners, leading to late diagnoses in the advanced stages of cancer [45]. Since dentists play a vital role in the screening and prevention of oral cancer [18], continuous education programs regarding oral cancer should be mandatory. It has been found that these programs proved to positively impact dentists’ knowledge and practices of oral cancer prevention and early detection [46,47]. Hopefully, the vast majority of the dentists in the present study were interested in joining continuous education courses related to oral cancer, similar to that reported in other studies [18,28,30,48].

Regarding oral cancer dental education, undergraduate dental students in Palestinian universities are sensitized to the topic of oral cancer from the third year of the curriculum. In this year, the oral pathology course focuses on the carcinogenesis of oral cancer, premalignant oral disorders, and the risk factors. This course consists of lectures, structured interactive sessions, and the assessment of histopathological slides. In the fourth and fifth years, oral medicine courses focus on the theoretical background of the diagnosis and treatment of malignant and potentially malignant oral lesions. These courses comprise lectures, structured interactive sessions, and seminars. Clinically, fourth- and fifth-year students are trained to examine any patient thoroughly prior to performing any dental treatment. In addition, fifth-year students have a mandatory oral medicine clinic where they are exposed to potentially malignant and oral cancer lesions, so as to practice examining, diagnosing, and managing similar cases [32]. However, more efforts should be made to review and enhance these curricula.

The present study gains its importance from being a beneficial inexpensive prompt cross-sectional analysis that shed light on an important part of dentists’ knowledge, opinions and practices. Another strength of this study is that this study targeted all practicing dentists in private, university and government sectors in Palestine. Therefore, we are now aware of the gaps that must be filled in order to tailor special continuous education programs, fulfilling this urgent need. We should also consider new educational methods and review the adequacy of the current dental curricula of oral cancer prevention in our dental schools. Furthermore, the authors are unaware of a previous questionnaire that has assessed oral cancer knowledge, opinions, and clinical practices among practicing dentists in Palestine. However, caution should be exercised when rendering the findings of this study due to some methodological limitations. It is a self-reported questionnaire, and the responses were subjective and may not reliably reflect the actual levels of knowledge, opinions and practices of the practicing dentists in Palestine. In addition, the relatively low response rate of 39.1% could present a non-response bias that may not allow us to generalize the results to all Palestinian practicing dentists.

## 6. Conclusions

With regard to the level of knowledge of oral cancer and its risk factors, about 70% of the surveyed Palestinian dentists had poor knowledge; while 84% had positive opinions related to oral cancer prevention and oral mucosal screening. Concerning the clinical practices of oral cancer prevention and oral mucosal screening, only 29.9% routinely screened their patients for mucosal lesions, attributing this to inadequate training and inadequate time for performing routine mucosal screening.

The data from this study highlight the need for reviewing the undergraduate/graduate dental curricula to address any knowledge and practice deficiencies, and to enhance the communication skills of future Palestinian dentists. Risk assessment and thorough head and neck cancer screening should become routine practices that are taught and trained in dental schools. In addition, currently licensed practitioners should undergo mandatory continuous educational programs on oral cancer. Furthermore, the importance of preventive behaviors, such as a reduction in the exposure to risk factors, and routine oral cancer screening should receive more attention.

## Figures and Tables

**Table 1 healthcare-11-01005-t001:** Demographic and professional data (*n* = 254).

Participants’ Data	N	%
**Age**		
>30	167	65.7
**Sex**		
Female	111	43.7
**Years of experience**		
≤5	85	33.5
6–15	108	42.5
>15	61	24.0
**Work setting**		
Private	205	80.7
University/Public	8	3.2
Both	41	16.1
**Specialty**		
General dentistry	203	79.9
Orthodontics/Pediatrics	12	4.7
Restorative/Endodontics	5	2.0
Prosthodontics	11	4.3
Oral surgery	12	4.7
Oral pathology	2	0.8
Periodontics	9	3.6

**Table 2 healthcare-11-01005-t002:** Number and percentage of participating dentists who correctly responded to the knowledge questions and who identified high risk factors for oral cancer (*n* = 254).

Question	Survey Results (Correct (%))
What is the most common oral cancer?	193 (76.0)
Excluding the lip, which of the following are the two most common sites of oral cancer?	95 (37.5)
The most common age of the patients to develop oral cancer is:	90 (35.4)
The symptom most commonly experienced by a patient with early oral cancer is:	79 (31.1)
The survival rate of oral cancer ranges between:	77 (30.3)
The mortality rate of oral cancer is greater among:	180 (70.9)
Early oral cancer lesions usually appear as a:	204 (80.3)
**Risk factor**	
Family history of oral cancer	26 (10.2)
Low consumption of fruits and vegetables	101 (39.8)
Poorly fitting dentures	65 (25.6)
Sun exposure (for lip cancer)	219 (86.2)
Older age	171 (67.3)
Human papilloma virus	190 (74.8)
Consumption of spicy food	126 (49.6)
Prior oral cancer lesion	236 (92.9)
Hot beverages and foods	167 (65.7)
Poor oral hygiene	79 (31.1)
Use of tobacco products	247 (97.2)
Consumption of alcohol	236 (92.9)
Obesity	49 (19.3)

**Table 3 healthcare-11-01005-t003:** Knowledge and opinions’ scores of participating dentists about oral cancer prevention and oral mucosal screening (*n* = 254).

Variable	N	%	Mean (SD)
Knowledge			11.14 (2.71)
Poor (0–12)	178	70.1	
Good (>12)	76	29.9	
Opinion			7.94 (1.49)
Negative (0–6)	40	15.7	
Positive (>6)	214	84.3	

**Table 4 healthcare-11-01005-t004:** Number and percentage of participating dentists who revealed positive opinions related to oral mucosal screening and oral cancer prevention.

Question	Survey Results (Positive Opinion (%))
I have sufficient knowledge concerning the prevention and diagnosis of oral cancer.	154 (60.6)
It is the role of the dentist to screen for oral mucosal pathology.	241 (94.9)
Screening of oral mucosal soft tissues should occur for all new patients.	210 (82.7)
Screening of oral mucosal soft tissues should occur for all recall patients.	189 (74.4)
I am comfortable discussing the presence of a suspicious oral mucosal lesion with my patient.	194 (76.4)
I can influence a patient to stop smoking.	126 (49.6)
I should provide my patients with smoking or alcohol cessation counseling and advice.	235 (92.5)
Patients could detect mucosal pathology on their own.	197 (77.6)
I should follow up patient referrals for pathology.	239 (94.1)
Would you like more information and continuing education regarding the prevention and early detection of oral cancer?	232 (91.3)

## Data Availability

The data that support the findings of this study are available from the corresponding author, upon reasonable request.

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
