# Peer review of "Knowledge, Opinions, and Practices of Oral Cancer Prevention among Palestinian Practicing Dentists: An Online Cross-Sectional Questionnaire"

_healthcare, 2023, doi:10.3390/healthcare11071005_

Round 1
Reviewer 1 Report
Dear Authors ,
Appreciate the efforts for the survey ,The study design is cross sectional which do not have background and the further recommendations .
Author Response
Appreciate the efforts for the survey. The study design is cross sectional which do not have background and the further recommendations.
Response: Thank you for this valuable comment.
Text Change: Added. L39-47, 324-331
Reviewer 2 Report
The authors of the current paper explored the awareness of oral cancer in Palestinian dentists. Prior to publication, there are some proposed revisions that may need to be done for improvement.
1. The title should be rephrased to reflect the main finding of the study.
2. Since the locality is Palestine, the introduction should contain information on the incidence and prevalence of oral cancer in Palestine. Moreover, this should also contain the health policies of Palestine pertaining to oral cancer. The last paragraph should exhaust the reasons why this study and its results are novel.
3. Under Materials and Methods, when the authors mentioned Google Drive as the software where the questionnaire was based, they might mean Google Forms that are saved in Google Drive. This needs to be clarified.
4. Since most of the data were presented as N (%). It is imperative to explain or mention Gaussian distribution in selecting parametric tests (i.e. t-test and ANOVA).
5. Figures 2, 3, and 4 may best be presented as tables to maintain uniformity with the previous tables. The p-value of the t-tests and ANOVA should also reflect in the appropriate tables. Reporting guidelines should also be used in reporting the statistics. For instance, it should contain the mean and SD or SEM, F, df, and sig.
6. Since much of the discussion is leaning on dental education and preparation of Palestinian dentists, there should be a short discussion on Palestinian dental education and the continuing dental education programs of the Palestine Dental Association or the Ministry of Health.
Author Response
The authors of the current paper explored the awareness of oral cancer in Palestinian dentists. Prior to publication, there are some proposed revisions that may need to be done for improvement.
- The title should be rephrased to reflect the main finding of the study.
Response: Thank you for this valuable comment,
Text Change: modified to “Knowledge, opinions, and practices of oral cancer prevention among Palestinian practicing dentists: an online cross‑sectional questionnaire”
- Since the locality is Palestine, the introduction should contain information on the incidence and prevalence of oral cancer in Palestine. Moreover, this should also contain the health policies of Palestine pertaining to oral cancer. The last paragraph should exhaust the reasons why this study and its results are novel.
Response: Thank you.
Text Change: Added. L 39-47, 79-87,
- Under Materials and Methods, when the authors mentioned Google Drive as the software where the questionnaire was based, they might mean Google Forms that are saved in Google Drive. This needs to be clarified.
Response: Thank you.
Text Change: Corrected.
- Since most of the data were presented as N (%). It is imperative to explain or mention Gaussian distribution in selecting parametric tests (i.e. t-test and ANOVA).
Response: Thank you.
Text Change: Added. L 138-141
- Figures 2, 3, and 4 may best be presented as tables to maintain uniformity with the previous tables. The p-value of the t-tests and ANOVA should also reflect in the appropriate tables. Reporting guidelines should also be used in reporting the statistics. For instance, it should contain the mean and SD or SEM, F, df, and sig.
Response: Thank you. I omitted all figures as they do not add more than mentioned in tables and text. P-value was included in the text each time we mention the effect of sex, years of experience and work setup on the level of the dentists’ knowledge, opinions, and clinical practice. SD was reported in Table 3.
Text Change: All figures are omitted.
- Since much of the discussion is leaning on dental education and preparation of Palestinian dentists, there should be a short discussion on Palestinian dental education and the continuing dental education programs of the Palestine Dental Association or the Ministry of Health
Response: Thank you for this valuable comment.
Text Change: Added. L298-310
Reviewer 3 Report
This article is satisfactory overall. You can go for publication.
Author Response
This article is satisfactory overall. You can go for publication.
Response: Thank you.
Reviewer 4 Report
The article presents the methodology and the results of a survey sent to Palestinian dentists in 2022. There is a need of major revisions before considering its publication in Healthcare. Please find my comments in the attached file.

Author Response
Reviewing: “Cross sectional questionnaire of oral cancer awareness among dentists practicing in Palestine” by Shadid and Hasbash.
The article presents the methodology and the results of a survey sent to Palestinian dentists in 2022. There is a need of major revisions before considering its publication in healthcare. Please find below my comments.
- A professional verification of the English language is needed.
Response: Thank you.
Text Change: Modified.
- Introduction, l.38-39: tobacco and alcohol are known to be the main risks factors for oral cancers worldwide, not only in Africa
Response: Thank you.
Text Change: Modified. L 48-54
- Introduction, l.44-46: the sentence repeats what was already explained previously. It should be simplified.
Response: Thank you.
Text Change: Modified. L 48-54
- Introduction, l.48: the word “demonstrate” is not adapted here.
Response: Thank you.
Text Change: Modified. L55-58
- Introduction, l.50: are these survival rates also 5-year survival rates?
Response: Yes
Text Change: Clarified. L55-58
- Introduction, l.52: the Global Oral Health is a report written by a group of experts. This is not clear in the manuscript.
Response: Thank you.
Text Change: Modified. L60
- Introduction, l.66-68: What are the conclusions of these studies? It is specified later in the discussion. If the authors want to mention these studies in the introduction, they have to develop. Or they can delete the mention.
Response: I appreciate your comment and I add a brief conclusion of those studies.
Text Change: Added. L73-78
- Introduction, l.69-79: this paragraph is unclear. One sentence extends on 9 lines. It must be divided in several sentences. Moreover, the authors must define clearly the main objective and the secondary objectives
Response: Thank you.
Text Change: Modified.L79-87
- Mat&Meth, l.85: it is interesting to report the study following the CHERRIES guidelines. However, if the authors announce they have followed these guidelines, they should respect the paragraphs’ titles as indicated in the grid.
Response: Thank you. We review the CHERRIES guidelines, and most of the items are respected in the article.
- Mat&Meth, l.87: a survey cannot be created on Google Drive but on Google Form. Please modify.
Response: Thank you.
Text Change: Modified.
- M&M, l. 94: It should be specified how the authors have use the surveys proposed in the studies #22, 23, 31, 33. They may have used exactly the same questionnaire, or mix all the questionnaires, etc.
Response: Thank you. The authors have mixed all of the questionnaires.
Text Change: Clarified. L105
- M&M, l. 95-96: from “content validity” to “work [31]” the sentence is not clear.
Response: Thank you.
Text Change: Modified. L107
- M&M, l.98: the word “slightly” is not precise. How many modifications were performed?
Response: The modification just involved rephrasing a few sentences to be clearer.
Text Change: Clarified. L109
- M&M, l.104-108: The paragraph must be written differently because it is difficult to understand.
Response: Thank you.
Text Change: Clarified. L113-117
- M&M, l.109-110: What is explained is that the authors have used 4-points Likert scales. They must justify their choice because today it is known that 3 or 5-points Likert scales better enables the respondent to express his doubts.
Response: I agree with you that Survey measurement scales preferably have at least five points. Presumably that provides enough room for variation, and it provides a neutral midpoint as well. However, this does not have sufficient scientific based evidence. Four-point scales still have advantages over five-point scales that it can be evenly split into simple dichotomies. We can report how many agree vs. how many disagree, this makes it easy to report one number simply and directly without misleading the audience. Does this distort the truth if people are forced into non-neutral choices? No, not necessarily. Many survey questions can be written so that everyone will or should have an opinion. Offering neutral options allows them to move on without giving careful thought to the question.
- M&M, l.121: the word “alpha” is repeated twice
Response: Thank you.
Text Change: Modified.
- M&M, l.122: the authors must place a reference at the end of the sentence when they affirm the coefficient suggests an adequate internal consistency.
Response: Thank you.
Text Change: Added. L132
- M&M, l.124: a Google Drive Excel document is traditionally called a Google Sheet.
Response: Thank you.
Text Change: Modified.
- M&M, l.125: SPSS is a software. It must be mentioned and the sentence written differently.
Response: Thank you.
Text Change: Modified. L134-135
- Table 1: it can be simplified with only one line for Age (>30) and only one line for Sex (Women). It is easy to understand that the inverse proportion is for younger and men.
Response: Thank you.
Text Change: Modified.
- Results, l.141: maybe the (12.71) is for standard deviation but it must be indicated. With a ± for example.
Response: Thank you.
Text Change: Modified. L152
- Table 2: the N (%) column should be deleted. This comment is also applicable to Table 4.
Response: Thank you.
Text Change: Modified.
- Table 2: it is difficult to understand what does the Yes mention in the first row mean. This comment is also applicable to Table 4.
Response: Thank you. In Table 2, “Yes” means a correct answer. In Table 4, “Yes” means a positive opinion.
Text Change: Clarified.
- Figure 1: the police is too small under the diagrams and difficult to read. Moreover, is it possible to indicate for each item which answer was considered as true?
Response: I omitted all figures as they do not add more than mentioned in tables and text.
Text Change: All figures are omitted.
- Results, l154: what is “a positive opinion”? Does it mean that the Palestinian dentists are in favor of prevention? Fortunately …
Response: Yes
- Figure 2: data labels must be written in a larger font inside the diagrams
Response: I omitted all figures as they do not add more than mentioned in tables and text.
Text Change: All figures are omitted.
- Results, l.177-181: I don’t understand the difference between the specialists mentioned in the two sentences.
Response: Thank you. The respondents were asked two different questions:
Have you ever referred to a specialist for a suspicious oral mucosal lesion?
Which type of health professional do/would you refer suspicious oral mucosal lesions to?
So the specialist in “74.4% had referred patients with a suspicious lesion to a specialist” is the response for the first question without determining the type of specialist they refer to; while the second one is specified.
- Figure 3: is it essential to report all the diagrams as their results are also mentioned in the text. This is also and especially true for the Figure 4 that does not bring anything new to the reader.
Response: I agree with you and omitted all figures as they do not add more than mentioned in tables and text.
Text Change: All figures are omitted.
- Results, l.200-206: The paragraph may be deleted as the Tables are already cited in text previously.
Response: I agree with you.
Text Change: Deleted.
- Discussion, l.211: a brief sum up of the initial hypothesis should be pertinent before saying it has been rejected.
Response: Thank you.
Text Change: Added. L200-202
- Discussion, l.216-219: I disagree with the analysis proposed by the authors. Indeed, it is not pertinent to compare the response rates between Palestinian dentists and for example Australian dentists. They are much more numerous in Australia, so the comparison of proportions is not interesting. The authors should mention the absolute values of participants.
Response: Thank you for your comment. I agree with you.
Text Change: Modified.
- Moreover, in the same paragraph, the conclusion “could be attributed to insufficient confidence some dentists feel toward oral cancer knowledge” is a bit of a quick shortcut. The participation rate, which is not so low in reality, may simply be related to the fact that dentists did not see the questionnaire or did not have time to answer it.
Response: Thank you for your comment, I agree with you.
Text Change: Modified.
- Discussion, l.263-264: Do the authors have information concerning the same knowledge of dentists concerning biopsy procedures? It seems to be essential in managing oral mucosal lesions.
Response: No.
- Discussion, l.317-318: it has already been mentioned several time in the manuscript that the present article may be the first to describe the knowledge of Palestinian dentists concerning oral cancers. Once is enough.
Response: Modified.
Reviewer 5 Report
The paper clearly identifies the gap between the oral practitioner and their knowledge regarding oral cancer. The paper is well designed but need few minor corrections.
1. Some sentences are too large and need to be broken into two.
2. Minor English corrections are needed especially in the discussion section.
3. Author needs to clearly outline the major outcome of this study regarding the results they get. The documented one can be presented in a more scientific manner.
Best of luck
Author Response
The paper clearly identifies the gap between the oral practitioner and their knowledge regarding oral cancer. The paper is well designed but need few minor corrections.
- Some sentences are too large and need to be broken into two.
Response: Thank you.
Text Change: Modified.
- Minor English corrections are needed especially in the discussion section.
Response: Thank you.
Text Change: Modified.
- Author needs to clearly outline the major outcome of this study regarding the results they get. The documented one can be presented in a more scientific manner.
Response: Thank you.
Text Change: Modified.
Round 2
Reviewer 2 Report
The authors have made substantial and pertinent changes to the manuscript based on the previous review.
Reviewer 4 Report
I want to highlight the work proposed by the authors during the time of reviewing. From my point of view, they have perfectly understood my initial comments and I have really appreciated the quality of their answers. Their paper is know suitable for publication in Healthcare.